# SIPA1 Is a Modulator of HGF/MET Induced Tumour Metastasis via the Regulation of Tight Junction-Based Cell to Cell Barrier Function

**DOI:** 10.3390/cancers13071747

**Published:** 2021-04-06

**Authors:** Chang Liu, Wenguo Jiang, Lijian Zhang, Rachel Hargest, Tracey A. Martin

**Affiliations:** 1Division of Cancer and Genetics, School of Medicine, Cardiff University, Heath Park, Cardiff CF14 4XN, UK; LiuC56@cardiff.ac.uk (C.L.); jiangw@cardiff.ac.uk (W.J.); 2Peking University School of Oncology and Peking University Cancer Hospital, Fucheng Road, Beijing 100142, China; lijzhang@yeah.net

**Keywords:** signal induced proliferation associated 1 (SIPA1), hepatocyte growth factor (HGF), N-methyl-N′-nitroso-guanidine human osteosarcoma transforming gene (MET), tight junction (TJ), non-small cell lung cancer (NSCLC)

## Abstract

**Simple Summary:**

The role of Signal Induced Proliferation Associated 1 (SIPA1) in lung cancer remains largely unknown. This study aimed to evaluate the importance of SIPA1 in the development and progression of lung cancer, to demonstrate the cellular functions of SIPA1 and the molecular mechanisms involved.

**Abstract:**

Background: Lung cancer is the leading cause of cancer death. SIPA1 is a mitogen induced GTPase activating protein (GAP) and may hamper cell cycle progression. SIPA1 has been shown to be involved in MET signaling and may contribute to tight junction (TJ) function and cancer metastasis. Methods: Human lung tumour cohorts were analyzed. In vitro cell function assays were performed after knock down of SIPA1 in lung cancer cells with/without treatment. Quantitative polymerase chain reaction (qPCR) and western blotting were performed to analyze expression of HGF (hepatocyte growth factor), MET, and their downstream markers. Immunohistochemistry (IHC) and immunofluorescence (IFC) staining were performed. Results: Higher expression of SIPA1 in lung tumours was associated with a poorer prognosis. Knockdown of SIPA1 decreased invasiveness and proliferation of in vitro cell lines, and the SIPA1 knockdown cells demonstrated leaky barriers. Knockdown of SIPA1 decreased tight junction-based barrier function by downregulating MET at the protein but not the transcript level, through silencing of Grb2, SOCS, and PKCμ (Protein kinase Cµ), reducing the internalization and recycling of MET. Elevated levels of SIPA1 protein are correlated with receptor tyrosine kinases (RTKs), especially HGF/MET and TJs. The regulation of HGF on barrier function and invasion required the presence of SIPA1. Conclusions: SIPA1 plays an essential role in lung tumourigenesis and metastasis. SIPA1 may be a diagnostic and prognostic predictive biomarker. SIPA1 may also be a potential therapeutic target for non-small cell lung cancer (NSCLC) patients with aberrant MET expression and drug resistance.

## 1. Introduction

Lung cancer is the commonest form of cancer and the leading cause of cancer death both in the UK and worldwide [1]. Non-small cell lung cancer (NSCLC) is the most prevalent type of lung cancer, accounting for approximately 85% of all lung cancers [2]. In comparison to small cell lung cancer (SCLC), NSCLC tumours contain numerous cancer driver gene mutations which may be targeted for treatment [3]. Multiple studies have shown that targeted therapies have improved the prognosis and survival of NSCLC patients carrying appropriate driver genes [4]. In developed countries, all NSCLC patients, regardless of their clinical characteristics, are routinely tested for suspected mutation points, such as epidermal growth factor receptor (EGFR) mutations, anaplastic lymphoma kinase (ALK) fusions, ROS proto-oncogene 1 (ROS1) fusions, and N-methyl-N′-nitroso-guanidine human osteosarcoma transforming gene (MET) mutation and amplification [5,6].

High expression of MET (a receptor tyrosine-protein kinase) can be observed in approximately 60% of NSCLC patients and correlates with poor prognosis [7]. In NSCLC, anomalous expression of MET may work as the driver of tumorigenesis and the cause for drug resistance to other therapies [8]. MET exon 14-skipping mutations and amplification of MET are the most frequent types of MET mutation found in NSCLC and are already included in the routine tests for driver oncogenes and therapeutic targets [5,6]. The *MET* gene encodes the protein MET which acts as the receptor for hepatocyte growth factor (HGF) [9]. MET belongs to the receptor tyrosine kinase (RTK) family with a conservative molecular structure and intracellular downstream signaling pathways [10]. Usually, HGF activation of MET acts in a paracrine pathway at the cell surface where there is a cycle of degradation and internalization. HGF/MET has been demonstrated to play an essential role in the ability of a tumour to metastasize by regulating cell motility, migration, proliferation and angiogenesis, and in addition, modulating the tight junction (TJs) cell to cell barrier [11,12]. TJ, together with anchoring (adherens) junctions and communicating (gap) junctions, constitute the epithelial cell junction, which is the basic structural entity which strengthens the mechanical connection between cells. They also maintain the integrity of tissue architecture and help coordinate the function of cells within human tissues [11]. TJ are located in the areas between the cell membranes of adjacent cells, and can completely occlude the space between cells and isolate the extracellular space, thereby creating an intercellular barrier which functions to maintain permeability and polarity, prevents migration and motility, mediates cell to cell adhesion, and transduce cell signaling which plays a role in differentiation and growth of epithelial and endothelial cells [13]. The TJ proteins consist of three major components: integral transmembrane proteins, peripheral or plaque anchoring proteins, and TJ associated or regulatory proteins. It has become increasingly recognized that human cancer is frequently associated with failure of epithelial cells to form TJ and to establish correct apicobasal polarity [14]. Changes in the expression and/or distribution of TJ proteins may result in complete loss of the TJ structure allowing cancer cell invasion and progression [11]. Therefore, TJs are an important factor in the progression of tumours. Our previous research has shown that TJ are largely regulated by the HGF/MET signaling pathway [15]. In human vascular endothelial cells (HUVECs), HGF treatment reduced the transendothelial resistance (TER) and increased paracellular permeability (PCP), which are regarded as key measurements of cell to cell TJ function [15,16,17]. HGF/MET signaling disrupted TJs in the breast cancer cell lines MDA-MB-231 and MCF-7 by downregulating the expression of several important TJ molecules, such as occludin, at both transcript and protein levels [18]. In prostate cancer, PC-3 cells had reduced TER and increased PCP after treatment with HGF [19]. Signal induced proliferation associated protein 1 (SIPA1), appears to play a crucial role in the regulation of TJs by HGF/MET.

SIPA1 was first cloned from murine cells in 1995 [20], and the human SIPA1 cDNA was identified in 1997 [21]. Initially, SIPA1 was believed to be a specific mitogen induced GTPase activating protein (GAP) for several Ras-related mediating proteins such Rap1 and Rap2 [20,21,22]. Subsequently, SIPA1 was found to be important in the development of various cancers and metastases. SIPA1 promotes the adhesion, migration and invasion of breast cancer MDA-MB-231 cells by binding to the promoter region of ITGB1 (integrin β1) gene and activating it, thus stimulating the downstream integrin mediated focal adhesion kinase/Protein kinase B/matrix metalloproteinase 9 (FAK/Akt-MMP9) pathway [23]. Higher expression levels of SIPA1 were associated with worse prognosis and increased incidence of metastases for prostate cancer (CaP) patients. SIPA1 has been found to intensify the invasion efficiency, but decrease the attachment ability, of CaP cells by down regulating bromodomain protein 4 (BRD4) and extra cellular matrix (ECM)-related gene expression [24]. In human oral squamous cell carcinoma (OSCC), SIPA1 interacting with BRD4 increases the level of matrix metalloproteinase 7 (MMP7), leading to increased invasiveness of the OSCC cells and worse prognosis for OSCC patients [25]. In colorectal cancer (CRC) cell lines HT115 and CaCO-2, knockdown of SIPA1 increased the invasion, adhesion, and migration potential of cancer cells compared to the control group, while the ability to proliferate was decreased [26]. It may be that SIPA1 can exert similar effects in cells with either high or low/knocked down of SIPA1 in different cancer types due to their tissue specificity, an area that requires further study. Moreover, SIPA1 is considered to be involved in many cancer types, including cervical cancer [27], gastric cancer [28], lung cancer [29,30], and melanoma [31], suggesting that SIPA1 may have an active role during progression and metastasis of cancer. While there has been little research focused on the modulating function of SIPA1 on the regulation of HGF/MET in TJs, previous work from the host laboratory indicated that in breast and prostate cancer cells, the presence of SIPA1 was required for the regulation of HGF/MET in TJs. Knockdown of SIPA1 silenced the effect of HGF on TJs in breast and prostate cancer cells [32,33]. However, there are limited studies investigating the function of SIPA1 in lung cancer [29,30] and the mechanism of SIPA1 in cancer development and metastasis remains largely unknown, particularly the role of SIPA1 in regulating the TJs and reacting to HGF signaling in lung cancer cells. This project, therefore, aimed to evaluate the interaction between SIPA1 and HGF/MET, as well as their influence on lung cancer in terms of molecular activation, cellular behaviour and clinical relevance.

## 2. Materials and Methods

### 2.1. Lung Cancer Tissue Samples

A clinical cohort consisting of lung tumours (*n* = 148) together with adjacent background tissues (*n* = 148) was obtained from patients at Peking University Cancer Hospital immediately after surgery and stored in a freezer at −80 °C until used. All protocol and procedures were approved by the Peking University Cancer Hospital Research Ethics Committee and informed consent was obtained from the patients.

### 2.2. Cell Culture

The cell lines used throughout this study to compare different types of lung cancer were the lung cancer cell lines A549 (an epithelial lung carcinoma cell), SK-MES-1 (a squamous cell carcinoma derived from a metastatic site) and COR-L23 (a tumourigenic lung large cell carcinoma). A549 and SK-MES-1 cell lines were purchased from the American Type Culture Collection (ATCC, LGC Standard, Salisbury, UK) and COR-L23 was purchased from the European Collection of Animal Cell Cultures (ECACC, Salisbury, UK). A549 and SK-MES-1 cells were cultured in Dulbecco’s Modified Eagle’s medium (DMEM), and COR-L23 lung cancer cells were cultured in GI1640 containing 2 mM glutamine. Medium was supplemented with heat inactivated fetal calf serum (FCS) (Sigma-Aldrich, Poole, Dorset, UK) and an antibiotic cocktail comprising penicillin, streptomycin and amphotericin B (Sigma-Aldrich).

### 2.3. Establishment of the Stable SIPA1 Knockdown Cell Lines

SIPA1 knockdown in A549, SK-MES-1 and COR-L23 cells was performed using hammerhead ribozyme transgenes. Anti-SIPA1 hammerhead ribozymes targeting the secondary structure of SIPA1 mRNA were designed using the Zuker NA mFold program, according to the protocol provided [34]. SIPA1 knockdown cells were designated SIPA1-KD, whereas plasmid control cells were designated pEF-CT.

The ribozymes were synthesized using touchdown polymerase chain reaction (PCR) and cloned into the pEF6/V5-His TOPO TA expression plasmid vector (Invitrogen, Loughborough, UK). Ribozyme transgenes and empty plasmids as control were transfected into the lung cancer cell lines using the Easyjet Plus electroporator (EquiBio, Kent, UK). Transfected cells for use in this projected were verified by selection with blasticidin.

### 2.4. RNA Extraction and PCR

RNA extraction was performed according to the Tri Reagent protocol (Sigma-Aldrich). RT-PCR was performed using the GoScriptTM Reverse Transcription System kit (Promega, Southampton, UK). The reverse transcription conditions were 25 °C for 5 min, 42 °C for 60 min and finally 70 °C for 15 min. GoTaq Green master mix (Promega) was subsequently used to amplify the target genes. The cycling conditions of the procedure were: denaturation at 94 °C for 5 min; 32 cycles of 94 °C for 20 s, 55 °C for 20 s and 72 °C for 15 s; with an extension phase at 72 °C for 10 min followed by storage at 4 °C until use. All the reactions were carried out in a 2720 Thermal Cycler (Applied Biosystems, Paisley, UK). Electrophoresis on agarose gel stained by SYBR Safe Gel Stain (Life Technologies, Paisley, UK) was used to separate the DNA fragments.

Quantitative PCR (qPCR) was performed using the Ampliflour™ Uniprimer™ Universal system (Intergen Company^®^, New York, NY, USA). The Ampliflour probe consists of a 3′ region specific to the Z-sequence (ACTGAACCTGACCGTACA) present on the target specific primers and a 5′ hairpin structure labelled with a fluorophore (FAM). The cycling conditions of the procedure were: 94 °C for 5 min; up to 100 cycles of 94 °C for 10 s, 55 °C for 35 s and 72 °C for 20 s. The fluorescent signal is detected to determine a threshold for quantification of genes amplified in the reaction. Copy number of a target transcript is determined using the cycle number of a reaction when its fluorescence signal reaches the threshold. The sequences for the primers used in this project are shown in Appendix A.

### 2.5. Protein Extraction and Western Blotting

Cells were washed, detached, and lysed using lysis buffer, centrifuged and the concentration of proteins was measured using Bio-Rad DC protein assay kit (Bio-Rad Laboratories, Hemel-Hempstead, UK). Protein detection was performed using the sodium dodecyl sulfate-polyacrylamide gel electrophoresis (SDS-PAGE) and antibody recognition. Information of the antibodies used in this project are shown in Appendix A.

### 2.6. Tissue Microarray (TMA) and Immunohistochemistry (IHC)

The lung disease spectrum tissue microarray slide LC1201(Biomax, Cheltenham, UK) was used in this project. IHC was performed using the VECTASTAIN^®^ Universal ABC Kit (Vector Laboratories Inc., Upper Heyford, UK). The TMA slide was dewaxed and rehydrated and treated with 30% hydrogen peroxide (H_2_O_2_) followed by microwaving for 20 min. After cooling, the slide was blocked with horse serum (1–2 drops) in 5 mL of 1× OptiMax Wash Buffer (BioGenex, Fremont, CA, USA) for 2 h. After washing, the slide was incubated with primary antibody (10 μg/mL) overnight at 4 °C. Following incubation with the corresponding secondary antibody, VECTASTAIN^®^ Universal ABC complex (Vector Laboratories Inc.) was added, then 3,3′-Diaminobenzidine (DAB) chromogen (Vector Laboratories Inc.) and the slide incubated in the dark. The target protein was indicated by brown colouration.

### 2.7. Immunofluorescence (IFC)

Cells were seeded into 8 or 16 well glass chamber. Cells were fixed, treated with a permeabilization buffer, blocked by horse serum and incubated with primary antibody overnight at 4 °C. 4′,6-Diamidino-2-phenylindole (DAPI), for nucleus staining, together with secondary antibodies tagged with either fluorescein isothiocyanate (FITC) or tetramethylrhodamine isothiocyanate (TRITC) were added. The procedures were carried out in the dark and slides stored at 4 °C. The slides were examined under a fluorescent microscope and photographs obtained using an Orca digital camera (Hamamatsu, Welwyn Garden City, UK).

### 2.8. Protein Array

Protein was extracted from A549 Pef-CT and SIPA1-KD cells. After quantification, protein samples were sent to Kinexus™ Bioinformatics (Kinexus Bioinformatics, Vancouver, BC, Canada). Data were analyzed using several parameters including Globally Normalized Signal Intensity, percentage (%) change from control (CFC) and the Z-scores.

### 2.9. Transendothelial Resistance (TER) Assay

Transwell inserts (Millicell, MerkMillipore, Watford, UK) with 0.4 μm pores were placed into the wells of a 24 well plate. Appropriately 5 × 10^4^ cells in 50 μL medium were seeded into inserts until confluent with 1ml medium in the well outside the insert. Resistance across the membrane was then measured in triplicate using the EVOM^2^ Epithelial Volt/Ohm Meter (World Precision Instruments, Hitchin, Hertfordshire, UK). Change in resistance was used as the measurement of the change in the TJ function of the cells.

### 2.10. Electric Cell-Substrate Impedance Sensing (ECIS) Assay

An ECIS instrument (Applied Biophysics Inc., Troy, NJ, USA) was used to asses changes in cell behaviour assay. The 96 well ECIS plate was connected to the Applied Bio Physics-ECIS Software V1.2.135 (Applied Biophysics Inc.), and then stabilized in 200 μL of serum free medium. Cells in appropriate density were seeded on the plate with at least six replicates. After cell seeding, the 96 well ECIS plate was connected to the ECIS station in the incubator. Resistance increases as the cells attach to the electrode and begin spreading and the resistance will continue to increase until the cells reach confluence.

### 2.11. In-Vitro Cell Proliferation Assay

Medium (200 μL containing appropriate cells (2 × 10^3^) was added to 96 well plates. These plates were incubated at 37 °C, with 5% carbon dioxide (CO_2_), for periods of 24, 48, 72, and 120 h.

After incubation, the medium was aspirated, and the cells were fixed with 4% formaldehyde for 20 min before being stained with 0.5% crystal violet for 15 min. The plate was then rinsed with tap water and left to air dry. The crystal violet was solubilized using 200 μL of 10% acetic acid, and the number of cells was assessed by measuring the absorbance of the resulting solution at 540 nm using a spectrophotometer (ELx800, BIO-TEK, Swindon, UK). The growth rate was calculated as a percentage, using the absorbance from the corresponding plate collected at 24 h as a baseline.

### 2.12. In-Vitro Transwell Matrigel Invasion Assays

Transwell inserts containing 8.0 μm pores (Millicell, Merck KGaA, Darmstadt, Germany) were placed into the wells of a 24 well plate. Inserts were coated with 100 μL of serum free medium containing 50 μg Matrigel^®^ Basement Membrane Matrix (Corning Incorporated, Flintshire, UK, stock concentration 0.5 μg/μL) and left to dry for 2 h at 55 °C. After rehydration, 2 × 10^3^ cells in 200 μL culture medium were seeded into each insert. Medium was then added to the lower chamber of each well. The cells were incubated with 5% CO_2_ at 37 °C for 72 h. After incubation, the Matrigel layer and the non-invasive cells were removed and the invasive cells were then fixed with 4% formalin and stained with 0.5% crystal violet. Stained cells were subsequently counted and photographed under the microscope, and the crystal violet was solubilized using 10% acetic acid to measure the absorbance.

### 2.13. Statistical Analysis

Statistical analysis was performed using the SPSS 26 (SPSS, Inc., Chicago, IL, USA), GraphPad Prism 8 (GraphPad Software, La Jolla, CA, USA), and Minitab 14 (Minitab Ltd., Coventry, UK). The cumulative survival curves were generated using Kaplan-Meier plots and analyzed using log-rank test. The two-tailed Student’s *t*-test or Mann–Whitney U test were used for two group comparisons depending on data parameters. For PCR and western blotting, band intensity was quantified using Image J software. Statistical significance was set at *p* < 0.05.

## 3. Results

### 3.1. The Expression Level of SIPA1 in Lung Cancer and Its Clinical Relevance

Transcript expression of SIPA1 in lung cancer tissue was determined in the cohort, collected at the Peking University Cancer Hospital, using qPCR. The clinical and pathological features of the patients, as well as their expression level of the SIPA1 transcript are summarized in Table 1.

It is clear that SIPA1 was significantly upregulated in lung tumour tissues compared with adjacent normal lung tissues (*p* = 0.0141) and paired normal lung tissues (*p* = 0.0358) (Figure 1A). The transcript level of SIPA1 in stage T-3 lung cancer was statistically significantly higher than that in stage T-1 and T-2 tumours. Comprehensive analysis of the TNM stage of the lung cancers, showed that TNM2 tumours had higher SIPA1 transcript levels than stage TNM1 tumours, and SIPA1 transcript levels in TNM2-3-4 stage was comprehensively higher than in the TNM1 tumours, but the difference was not statistically significant (Figure 1B). In terms of the substage, SIPA1 levels in TNM 1A and 1B tumours were lower than those in TNM 2B which had increased metastasis; and the difference was statistically significant (Figure 1B).

Kaplan-Meier survival analysis revealed that patients from the Peking cohort with a high expression level of SIPA1 had lower overall survival compared to those patients with a low expression level of SIPA1. However, the data did not reach statistical significance (*p* = 0.204). We also analysed the distribution and clinical parameters of SIPA1 using a publicly available database from the Kaplan Meier plotter website (http://kmplot.com/analysis/index.php?p=background, accessed on 19 January 2019). This showed that high expression of SIPA1 was statistically correlated with worse post-progression survival (PPS) (*p* = 0.02) in all of the lung cancer patients (Figure 1C), while there was no statistically significant correlation of SIPA1 expression level with overall survival (OS) (*p* = 0.46) or progression-free survival (PFS) (*p* = 0.071) (data not shown).

With regard to the main histological types of lung cancer, the results from this cohort showed that in patients with lung adenocarcinoma, those with a low level of SIPA1 tended to have longer survival. Survival data of patients with lung adenocarcinoma cancers from the Kaplan Meier plotter website showed high SIPA1 level was accompanied by a poor outcome from cancer in terms of both OS (*p* = 0.0021) and PFS (*p* = 0.0065) (Figure 1D). However in patients from the Peking cohort with squamous cell carcinoma, those with a low level of SIPA1 tended toward a worse prognosis without statistical significance; and data from the Kaplan Meier plotter website also lacked statistical significance (Figure 1E).

To evaluate the protein level of SIPA1 at different stages of lung cancer, IHC was carried out and scored using a lung cancer tissue microarray (TMA, LC1201) (Figure 2). The IHC scoring did not show a significant difference of SIPA1 staining in lung tumours compared with all normal pneumonic tissue, neither did it show differences with normal tissue and adjacent normal tissue (data not shown).

### 3.2. Knockdown of SIPA1 Decreased the Proliferation and Invasion Potential of Lung Cancer Cells

PCR, qPCR and western blot were performed to detect the knockdown of SIPA1 at both mRNA and protein levels. SIPA1 was successfully knocked down at both mRNA (Figure 3A,B) and protein levels (Figure 3C) in A549, SKMES1 and CORL23 cell lines.

The in vitro growth assay indicated that SIPA1 knockdown significantly decreased the proliferation potential of the A549 cells, both after 3 days (*p* = 0.0001) and 5 days (*p* = 0.0020) (Figure 3D). The in vitro trans-well invasion assay showed the invasion potential was markedly reduced after knockdown of SIPA1 in A549 and SKMES1 lung cancer cell lines (Figure 3E).

### 3.3. Knockdown of SIPA1 Enhanced the Tight Junction Based Barrier Function of Lung Cancer Cells

The data from the ECIS assay showed that knockdown of SIPA1 in A549 cells enhanced cell-cell barrier function significantly, which was also demonstrated by the quantitative resistance analysis at time point 2.5 h, as well as the 3D figure, where we detected the resistance under various electric current frequencies. (Figure 3F). The TER experiment results showed similar findings: knockdown of SIPA1 enhanced cell-cell TJ function compared to control in the A549 cell line (Figure 3G).

### 3.4. TJ Markers Were Influenced by SIPA1 in Lung Cancer Cells

QPCR was performed in order to investigate gene expression of the major TJ proteins. In the A549 cell line, the effects of knockdown of SIPA1 showed that zonula occludens 1 (ZO1) and zonula occludens 3 (ZO3) were upregulated to some extent. Interestingly, levels of the junctional adhesion molecule (JAM) family members, whilst not changed at the transcriptional level appeared decreased at the protein level. Whilst transcriptional levels can be important indicators of changes in protein expression, the relationship is not always linear and changes in gene expression level are frequently not reflected at the protein level. Moreover, JAMs, as transmembrane proteins are often cleaved from the cell and thus not picked up during western blotting. This is an interesting finding that could be further investigated in the future. Afadin (AF6) was downregulated, whilst multi-PDZ domain protein 1 (MUPP1) was downregulated significantly; both of these proteins contain a PDZ domain. MARVEL domain-containing protein 3 (Marvel D3), which belongs to the TJ-associated MARVEL domain containing protein (TAMP) family was significantly downregulated by knockdown of SIPA1, whilst occludin, another TAMP protein, was upregulated by knockdown of SIPA1 at the transcript level.

The claudin family members 1, 2, 7, 8, 11, 14, and 17 were significantly downregulated and claudin 5 was significantly upregulated in SIPA1 knockdown A549 cells (Figure 4A). There are often poor correlations generally reported between the level of mRNA and the level of protein which may not be mutually exclusive. This disparity can be due to complicated and varied post-transcriptional mechanisms involved, moreover, protein half-live can differ significantly. The claudins constitute an important transmembrane TJ protein family. In SKMES1 cells, knockdown of SIPA1 upregulated the transcript level of ZO family proteins significantly, sharing the same trend as that found in A549 cells (albeit not statistically significant in A549 cells). We also found that AF6 was downregulated and MUPP1 was upregulated in SKMES1 cells. Both Marvel D3 and occludin, transmembrane proteins belonging to the TAMP family, were significantly upregulated by knockdown of SIPA1 at the transcript level (Figure 4B). Further work is required to establish the significance of these effects on the function of TJs in lung cancer.

Western blotting was performed to further investigate the potential relationship between SIPA1 and TJ components in A549 cells. Claudin7, claudin10 and claudin15 protein levels were decreased by knockdown of SIPA1 in A549 cells, although there appeared to be a very small non-statistically significant increase at the messenger level. The effect of knockdown of SIPA1 on the JAM family, occludin and PDZ domain containing proteins was not significant (Figure 4C).

Immunofluorescence staining was carried out to further investigate the visible impact of SIPA1 on cell morphology and tight junctions. ZO1 was chosen to stain in the A549 pEF-CT cells and SIPA1-KD cells. The immunofluorescence staining demonstrated that the intensity of ZO1 was strong in the pEF control cells but weak in the SIPA1 knockdown cells (Figure 4D); the graph illustrates this difference in intensity between pEF-CT and SIPA1-KD cells (using the two representative images, *n* = 20), *p* < 0.005. The apparent decrease in ZO1 intensity at the cell membrane in SIPA1-KD cells may indicate that it has been relocated from the TJ towards the perinuclear region, despite increased expression seen at the transcriptional and gross protein levels.

### 3.5. SIPA1 Was Correlated with HGF/MET Signalling Pathway and Tight Junctions

Analysis of the data from the Peking cohort revealed that SIPA1 is highly correlated with transcript levels of HGF, MET and Protein Tyrosine Kinase 2 (PTK2), or focal adhesion kinase (FAK), as well as with those of some TJ molecules such as JAM2, Marvel D2, Marvel D3, nectin2 and some TJ-regulating proteins such as RHOC, ROCK1 and ROCK2. However, the correlation between SIPA1 and macrophage stimulating 1 (MST1), or HGF-like protein (HGFL), and receptor of MST1 (MST1R or RON) was not significant (Figure 5A). Analysis based on the TCGA-LUAD database (available from the website https://www.cancer.gov/about-nci/organization/ccg/research/structural-genomics/tcga, accessed on 19 January 2019) showed hepatocyte growth factor activator (HGFAC, HGF activator), HGF, MET, MST1/MST1R, ERF/EGFR, PKT2 and its antagonist paxillin (PXN), together with PRKCA/PTEN, the regulator of PKT2, and SRC, the downstream protein of PKT2, were all correlated with SIPA1 at the transcriptional level with statistical significance (Figure 5B). Correlation analysis on the TCGA-LUAD database also demonstrated that TJ markers such as the gene F11R (encoding JAM1 protein), the ZO family encoding genes, Claudin proteins encoding genes such as claudin4, claudin5, claudin12, claudin15, and Marvel D3 were all highly correlated with SIPA1 at the transcriptional level (Figure 5C).

The phosphorylation status between A549 pEF-CT cells and the A549 SIPA1-KD cells was compared in order to determine if protein within the pathway could still be phosphorylated/dephosphorylated to identify potential downstream partner proteins. Those markers with the greatest increase or decrease in phosphorylation status after SIPA1 knockdown in A549 cells, indicated potential interaction markers with intracellular SIPA1. The website Reactome (www.reactome.org, accessed on 19 January 2019) was used to determine the key pathways involved and the most downregulated and upregulated markers after SIPA1 knockdown could be integrated to centralized signaling pathways (Appendix A and Figure 5D).The genome-wide overview schematic diagram of the signaling pathways was generated to show the overall review of the main area in which the pathways impacted by SIPA1 are located.

### 3.6. SIPA1 Was Involved in the Regulation of HGF/MET on TJs in Lung Cancer Cells

The in vitro trans-well invasion assay showed that invasion potential was markedly increased after the treatment with HGF on the pEF-CT A549 cells. After knockdown of SIPA1 in A549 cells, the invasion potential in reaction to treatment with HGF was reduced (Figure 6A). ECIS assay showed that HGF reduced the barrier function of pEF-CT A549 cells, particularly evident at 15h (where knockdown cells show no reduction in resistance when treated with HGF) and an inhibitor of MET (Crizotinib) could counteract the effect of HGF on pEF-CT cells. However, HGF did not reduce barrier function after knockdown of SIPA1 in A549 cells (Figure 6B). ECIS barrier function is taken as a surrogate marker of TJ function.

### 3.7. SIPA1 Regulates MET at the Protein Level, by Regulating the Internalization and Reuse of MET

RT-PCR and qPCR showed that the transcript level of MET was not influenced by knockdown of SIPA1 in A549 lung cancer cells. In SK-MES1 lung cancer cells, qPCR showed the transcript level of MET was actually higher in SIPA1-KD cells compared to pEF-CT cells (Figure 6C,D). Western blotting results showed that the protein level of MET was reduced in SIPA1 knockdown A549 and SKMES1 cells (Figure 6E). The phosphorylation of MET was also reduced in SIPA1 knockdown cells, but then so was total MET expression (Figure 6F). It appears that SIPA-KD results in reduced ZO1 expression and associated phosphorylation. This may indicate a key mechanism whereby SIPA1 is able to control the tightness of the TJ by effecting changes in ZO1 expression and switching off activity.

Western blotting showed the expression of PKCμ, Grb2 was downregulated by SIPA1 knockdown in A549 and SKMES1 cells (Figure 7A,B). Moreover, the expression of most SOCS family (SOCS1, 3, 5, 7), which can control HGF controlled growth was inhibited by SIPA1 knockdown in A549 and SKMES1 cells (Figure 7C).

## 4. Discussion

Lung cancer is the leading cause of malignant tumour death, independently of country or region [34]. Extensive evidence has suggested that the HGF/MET signaling pathway is essential in lung cancer tumorigenesis and progression via the alteration of cell apoptosis, growth, migration and morphology [35]. MET has been proposed as one of the driving oncogenes in lung cancer and the cause of drug resistance to EGFR TKIs [8]. Meanwhile, an increasing number of studies indicate the importance of TJs in cancer metastasis [11]. SIPA1 has been shown to work as a driving factor in a variety of tumours [22]. The potential of SIPA1 in regulating TJs has also been confirmed in breast cancer and prostate cancer in the host laboratory. The effect of the HGF/MET signaling pathway on breast cancer cells requires the presence of SIPA1 [19,32,33]. During this study a series of functional assays were performed in order to find out more about the role of SIPA1 in the function of lung cancer cells.

This study showed that knockdown of SIPA1 reduced the response of lung cancer cells to HGF in terms of invasion and barrier function. HGF enhanced the invasion of pEF control cells, while the enhancement could be blocked by SIPA1 knockdown. HGF decreased the barrier function of lung cancer cells, which was counteracted by SIPA1 knockdown with similarity to the small molecule MET targeted inhibitor, Crizotinib. Since knockdown of SIPA1 has a similar effect to pharmacological MET inhibition on blocking the regulation of HGF signaling on tight junctions, it is reasonable to assume that SIPA1 could have some potential interaction with MET.

The mechanism of interaction between SIPA1 and MET is an important issue to be determined in this study. SIPA1 gene expression was studied in the Peking lung cancer cohort and the TCGA database. The results showed that SIPA1 expression is highly correlated with some key markers in the HGF/MET signaling pathways and with TJ components. At the same time, protein phosphorylation array revealed that SIPA1 is involved in the regulation of complex signal transduction and RTK activation. This confirms what has previously been described-that SIPA1 and MET interact to effect changes in cell barrier function. It is known that the regulation of RTKs is an extremely complicated process involving multiple signaling pathways. Although protein array analysis has shown that there exists a co-relationship between SIPA1 and the RTKs family, to which MET belongs, the specific mechanism of interaction has yet to be demonstrated. However, the protein expression level of MET was suppressed by SIPA1 knockdown, as detected by western blotting; we can infer from this that the cell surface therefor lacked sufficient MET receptors to transmit extracellular HGF signals. Moreover, reduced MET expression as effected by SIPA1 knockdown led to the downregulation in MET phosphorylation (Tyr 1349 and 1365). The phosphorylation status of key sites of the MET receptor’s domain used to anchor downstream molecules was inhibited, limiting the function of MET to down-stream transmission signals.

MET regulation involves multiple different processes from transcription to degradation, which include MET oncogene mutations, MET gene methylation, transcription factors regulation, alternative splicing, microRNA regulation, MET translational regulation, proteolysis of MET, glycosylation and phosphorylation on MET, internalization, degradation and recycling of MET, nuclear localization of MET, and autoregulation of MET [36,37]. This project has shown that SIPA1 can influence the expression of MET, but the specific process (or processes) by which SIPA1 acts is still an area for future study. Some key regulatory molecules during the MET recycling process, such as Grb2, SOCS, and PKCμ, were selected for further analysis. Western blot results for these molecules showed that the protein expression of these molecules decreased after SIPA1 knockdown. In previous publications, Grb2, SOCS, and PKCμ all promoted internalization of MET, leading the process towards recycling to the membrane rather than degradation [37]. Knockdown of SIPA1 decreased the expression of these molecules which could be a potential approach to regulate MET receptors. Therefore, we can conclude that in lung cancer cells, Grb2, SCOS and PKCμ are the potential target molecules for SIPA1, thus promoting the recycling of MET, in order to maintain the MET receptor at a high level, thereby enabling the transmission of the HGF signal within the cells.

The findings of this study suggest that the expression of SIPA1 in lung cancer cells is closely related to the expression of TJ molecules and the cell’s barrier function. Gene expression correlation analysis showed the potential regulatory effect of SIPA1 on TJs. Data from the Peking cohort and the TGCA database (available from the website https://www.cancer.gov/about-nci/organization/ccg/research/structural-genomics/tcga, accessed on 19 January 2019) were analyzed. The components of TJ proteins that are highly correlated with expression of SIPA1 include JAM1, JAM2, ZO family, occludin, Marvel D2, Marvel D3, nectin2, and claudin4, claudin7, claudin12, and claudin15, which are proven in at least one database from the Peking cohort and/or the TGCA database. qPCR and/or western blotting demonstrated that claudins 1, 2, 7, 8, 11, 14 and 17 from the claudin family, and MUPP1 which contains a PDZ domain were downregulated after knockdown of SIPA1. Claudin5, and the PDZ domain containing protein AF6 was downregulated and ZO family proteins were upregulated in SIPA1 knockdown cells compared to pEF control cells. Previous studies have shown that SIPA1 contains a PDZ domain [38], as do other TJ markers such as the ZO family, AF6, MAGI1, MAGI2, MAGI3, PAR3, PAR6 and MUPP1. The common feature that SIPA1, AF6, and the ZO family all contain the PDZ domain, suggests that this could be a potential interaction site. It is possible that SIPA1 could interact with AF6 and the ZO family in the regulation of TJ function.

The transcriptional expression level of SIPA1 was found to be higher in cancer than adjacent normal tissue or normal tissue in the Peking clinical cohort. The IHC staining score of the lung tumour TMA lacks statistical significance. The protein level of SIPA1 in tumour tissue samples was higher than that in normal lung tissue and adjacent normal lung tissue, but the difference was not statistically significant (data not shown). There may be inconsistencies between SIPA1 gene transcripts and SIPA1 protein analysis. As previously discussed, there is a possibility that difference exists between the transcription level of SIPA1 gene and the protein level of translation, but the Peking clinical cohort had a larger sample number than the TMA cohort, and importantly, it has a higher number of normal tissues. The clinical cohort samples are fresh frozen from surgery and it was easy to quantify the expression of SIPA1, whilst the TMA samples are could only be used for qualitative or semi-quantitative analysis. In terms of tumorigenesis, the Peking clinical cohort data showed that the transcription level of SIPA1 was significantly higher in the advanced T stage and TNM stage tumours. From the prognostic analysis of lung cancer database, the median survival time of patients with high SIPA1 expression was significantly shorter than that of patients with low SIPA1 expression, and the difference is particularly significant in patients with adenocarcinoma. Transcription of SIPA1 was high in lung cancer tumour and more advanced TNM stages of lung cancer, and the high level of SIPA1 correlated with the worst prognosis of lung cancer patients, which all indicate that SIPA1 can be used as a potential independent clinical biomarker to predict the prognosis and recurrence of lung cancer. MET has been reported as a driving factor and therapeutic target for NSCLC. Crizotinib (Xalkori^®^, Pfizer, Surrey, UK) is approved by the FDA as a MET targeted inhibitor used in NSCLC patients with MET amplification and/or exon 14 skipping mutation [8]. Since SIPA1 can interact with MET, and SIPA1 can regulate the MET receptor protein, and the inhibition of SIPA1 has similar effects on the TJs of lung cancer cells as the MET inhibitor Crizotinib, it is possible that SIPA1 may be a potential therapeutic target in NSCLC and a potential substitute target for MET targeted therapy resistance in NSCLC.

## 5. Conclusions

In conclusion, SIPA1 plays an essential role in NSCLC tumorigenesis and metastasis, by enhancing invasion and proliferation and suppressing the barrier function of lung cancer cells. This is achieved via promoting Grb2, SOCS and PKCμ, ensuring the recycling of MET, and further ensuring normal transmission of the HGF/MET signaling pathway. Figure 8 summarizes the findings of this project: SIPA1 and HGF form an axis that is able to regulate the function of TJ’s in human lung cancer cells. This novel discovery offers a pathway that may be explored both biologically and clinically. Biologically, this of interest as there has previously been little information regarding the exact pathway involved in HGF effected changes of TJ function. Clinically, this pathway may provide possible routes for therapeutic intervention. How this can be achieved is beyond the scope of the current work and remains an important area of future research.

There now remains further work to be carried out. Since SIPA1 can interact with MET, the interaction with other RKTs family members, especially the crucial EGF/EGFR signaling pathway in NSCLC is worth investigation. The other mechanisms by which SIPA1 regulates the tight junctions of lung cancer cells require further exploration. A targeted inhibitor of SIPA1 protein needs to be developed and examined for its effect in vitro and In vivo prior to possible clinical application.

## Figures and Tables

**Figure 1 cancers-13-01747-f001:**
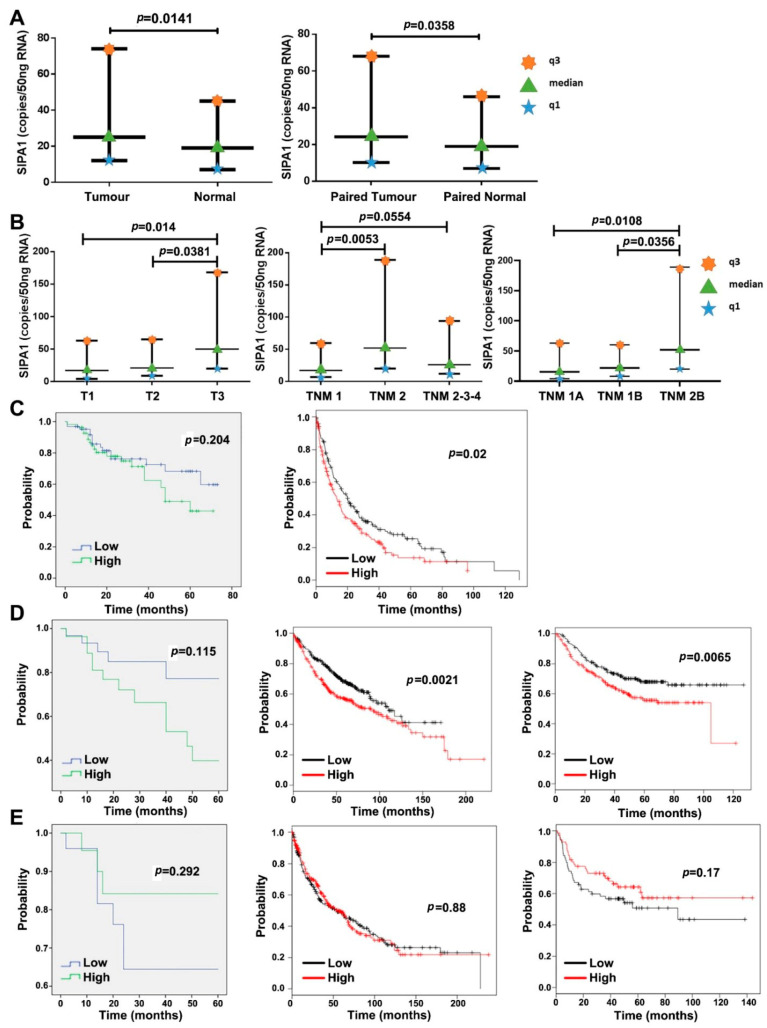
Expression level of SIPA1 in lung cancer and its clinical relevance. (**A**) Transcript level of SIPA1 in lung cancer and adjacent normal lung tissue (left) and paired adjacent normal lung tissue (right). (**B**) Transcript level of SIPA1 in different T stage, TNM stage and TNM substage lung cancer. (A&B-SIPA1 level was evaluated by copy number/50 ng RNA, and Mann-Whitney U test was selected to determine significance).(**C**) Kaplan-Meier survival models of correlation between SIPA1 transcript levels and survival of all lung cancer patients from Peking cohort-Overall survival (OS) (left), and post progression survival (PPS) (right). (**D**) Kaplan-Meier survival model of correlation between SIPA1 transcript levels and survival of lung adenocarcinoma patients. Left: SIPA1 transcript levels with OS in patients from Peking cohort; middle: SIPA1 transcript levels with OS in patients from KMplot website; right: SIPA1 transcript levels with PFS in patients from KMplot website. (**E**) Kaplan-Meier survival model of correlation between SIPA1 transcript levels and survival of lung squamous cell carcinoma patients. Left: SIPA1 transcript levels with OS in patients from Peking cohort; middle: SIPA1 transcript levels with OS in patients from KMplot website; right: SIPA1 transcript levels with PFS in patients from KMplot website.

**Figure 2 cancers-13-01747-f002:**
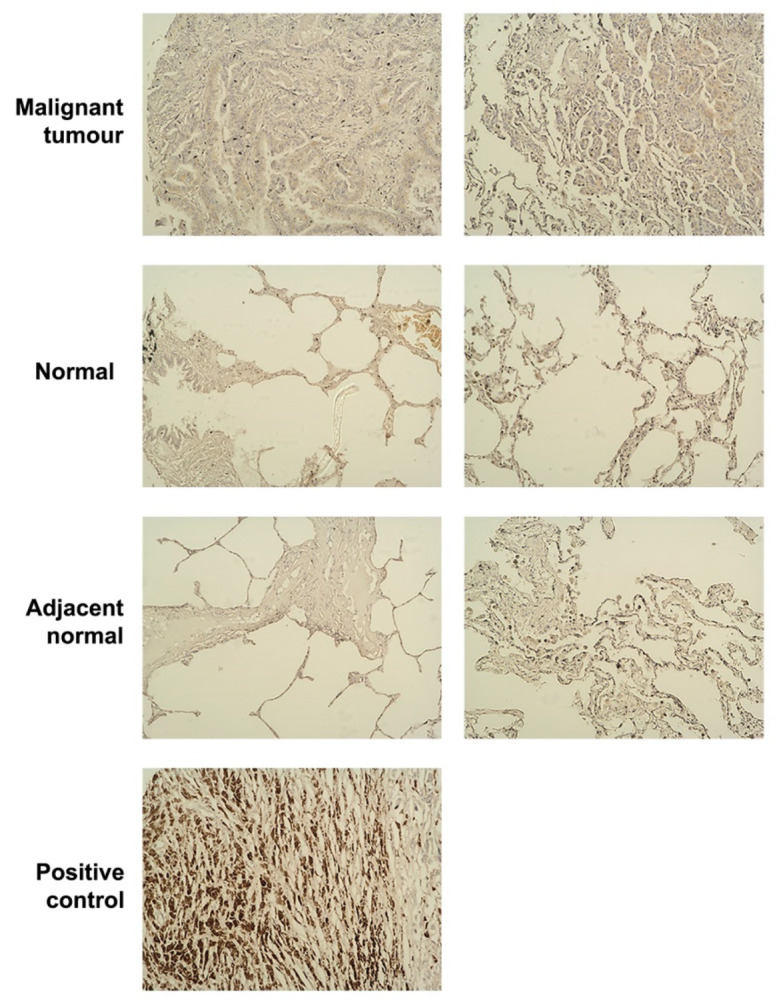
Representative IHC SIPA1 staining images in the TMA. Figure shows malignant tumour tissues, normal pneumonic tissues, cancer adjacent normal pneumonic tissues and the positive control from skin malignant melanoma. All the images were captured at 200× objective magnification.

**Figure 3 cancers-13-01747-f003:**
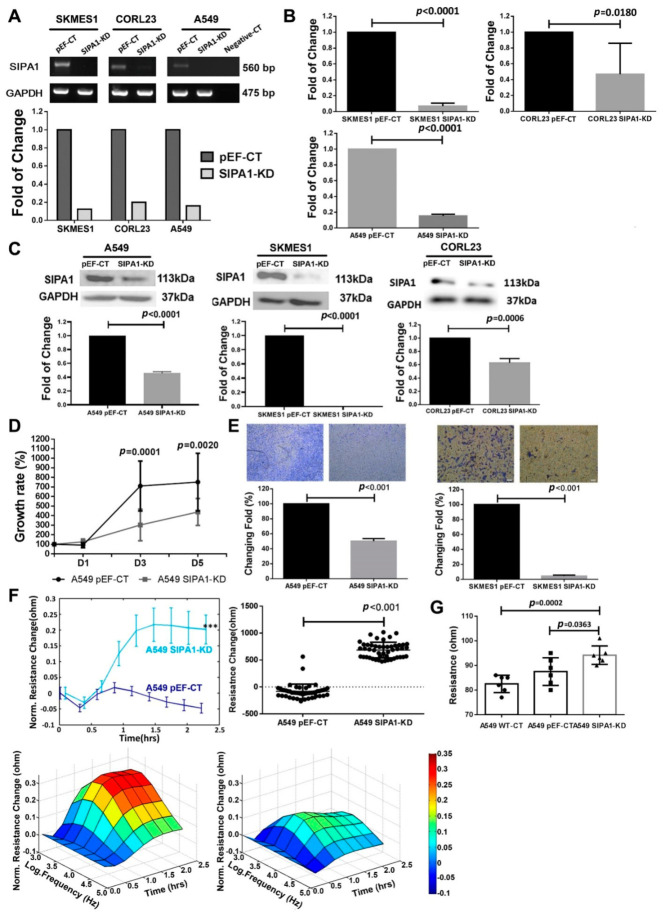
Knockdown of SIPA1 and its effect on the aggressive behaviour and barrier function of lung cancer cells. Verification of the knockdown of SIPA1 in SKMES1, CORL23 and A549 cell lines at mRNA level using PCR (**A**), qPCR (**B**) and at protein level using western blotting (**C**). (**D**) Knockdown of SIPA1 decreased the growth of the A549 cells significantly. Error bars show SD. (**E**) Knockdown of SIPA1 decreased the invasion of the A549 and SKMES1 cells significantly. (**F**) Knockdown of SIPA1 increased the resistance of the monolayer of the A549 cells significantly (upper left); quantitative analysis of the resistance at time point 2.5 h showed resistance (and TJ barrier function) of A549 cells was upregulated (upper right); recording the resistance of the cells under different electric current frequency to draw the 3D figure showed knockdown of SIPA1 increased the cell-cell tight junction barrier function (lower). (**G**) Changes in Transepithelial Resistance (TER) due to SIPA1 knockdown in A549 cells. Experiments were repeated *n* = 3. Asterisks indicate statistical significance.

**Figure 4 cancers-13-01747-f004:**
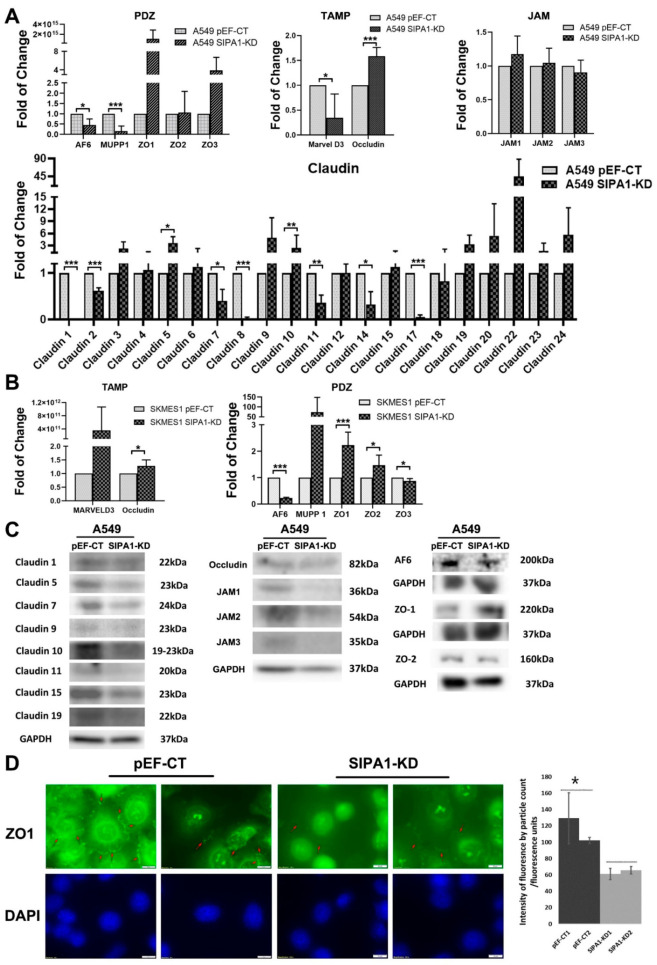
The impact of SIPA1 knockdown on tight junction molecules in lung cancer cells. (**A**) The effect of SIPA1 knockdown on gene expression of PDZ domain containing proteins, TAMP proteins, JAM family, and Claudin family in A549 cells. (**B**) The effect of SIPA1 knockdown on gene expression of TAMP proteins, and TJ proteins containing a PDZ domain in SKMES1 cells. (**C)** The effect of SIPA1 knockdown on expression of Claudin family, Occludin, JAM family, and PDZ domain containing proteins in A549 cell line. (**D**) Immunofluorescence staining for ZO1 in A549 pEF-CT cells and SIPA1-KD cells, accompanying graph shows significant difference between expression levels of ZO1 in pEF-CT cells (image 1and 2) versus SIPA1-KD (1 and 2). Experiments were repeated *n* = 3. *, **, *** indicate statistical significance.

**Figure 5 cancers-13-01747-f005:**
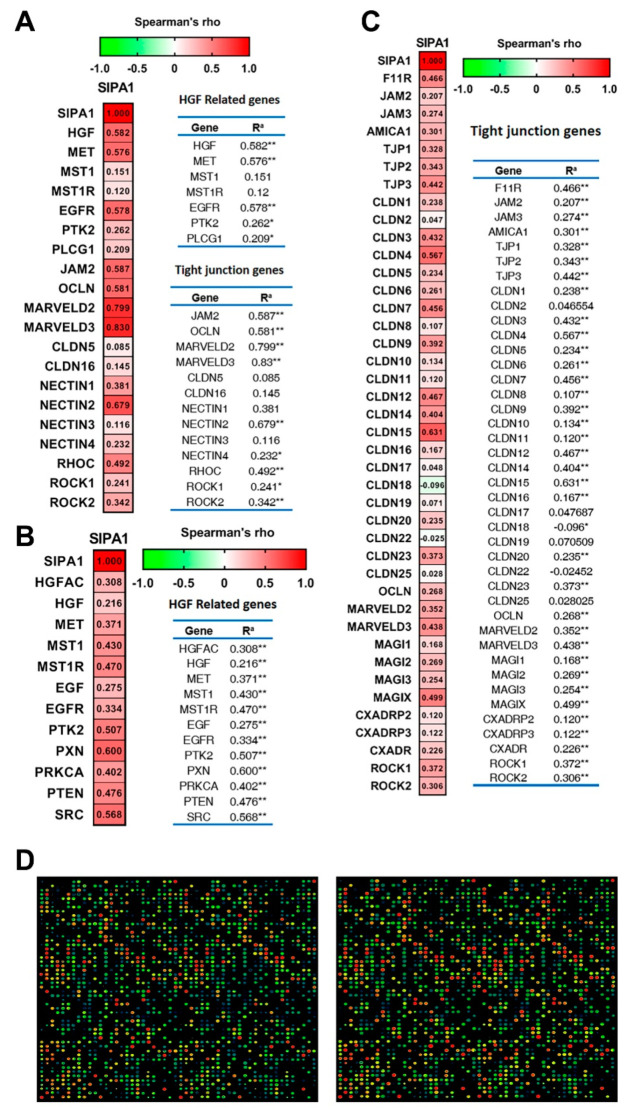
Correlation coefficient of SIPA1 with key HGF/MET components and tight junction markers. (**A–C**) show the Spearman’s rank correlation coefficient between SIPA1, HGF related genes and tight junction component genes analyzed from the Peking lung cancer cohort (**A**), and Spearman’s rank correlation coefficient between SIPA1 and HGF related genes (**B**), tight junction component genes (**C**) analyzed from TCGA-LUAD database. (**D**) Overall images of a direct fluorescent-dye labelled KAM-1325 antibody microarray slide detecting the protein sample from the A549 pEF control cells (left) and A549 SIPA1 knockdown cells (right). * and ** denotes statistical significance, i.e., *p* values below 0.05.

**Figure 6 cancers-13-01747-f006:**
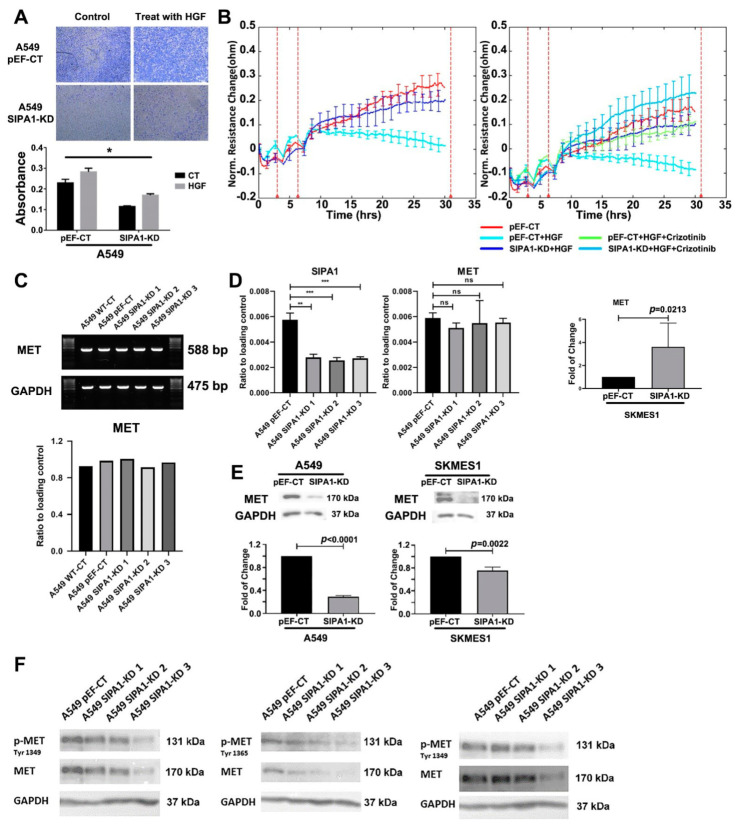
SIPA1 and HGF/MET signaling on the regulation of TJs in lung cancer cells. (**A**) HGF increased the invasion potential of pEF-CT A549 cells, and knockdown of SIPA1 decreased invasion in A549 pEF cells with/without the treatment of HGF. (**B**) HGF decreased resistance of A549 pEF-CT cells significantly but did not decrease resistance of A549 SIPA1 knockdown cells (left), indicating the regulation of HGF on TJs requires the assistance of SIPA1. The effect that HGF decreased resistance of A549 pEF-CT cells instead of A549 SIPA1-KD cells produced a similar result as that achieved when using the MET inhibitor (Crizotinib), and the combination of SIPA1 knockdown and MET inhibitor could further prevent regulation of HGF on A549 cells (right), all of which revealed that silencing SIPA1 has the same cellular influence on A549 cells as blocking HGF/MET signaling by Crizotinib. (**C**,**D**) The mRNA level of MET was not regulated by the knockdown of SIPA1 in lung cancer cell lines examined using PCR (**C**) and QPCR (**D**). (**E**) Western blot showing the protein level of MET was down regulated by knockdown of SIPA1 in A549 and SKMES1 lung cancer cell lines. (**F**). The phosphorylation level of MET was down regulated by knockdown of SIPA1 in A549 lung cancer cell lines which would be due to the reduction in total MET protein expression. Experiments were repeated *n* = 3. *, **, *** indicate statistical significance.

**Figure 7 cancers-13-01747-f007:**
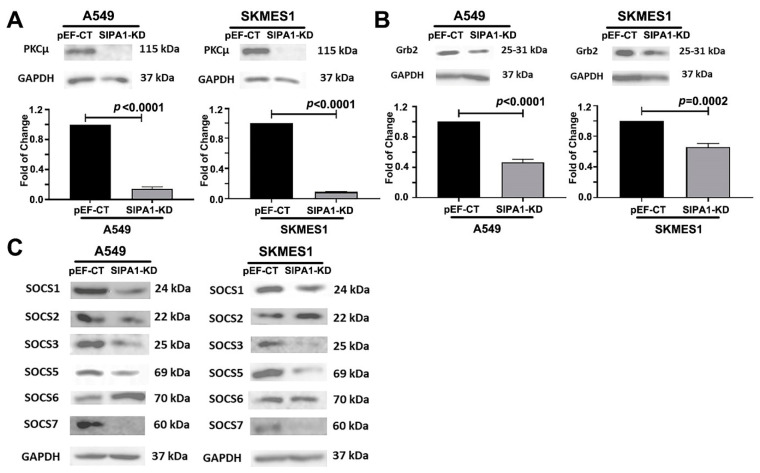
The key molecules regulated by the interaction of SIPA1 with HGF/MET signaling pathway in human lung cancer cells. The impact of knockdown of SIPA1 on the protein level of PKCμ (**A**), Grb2 (**B**), and SOCS family (**C**) in A549 and SKMES1 lung cancer cell lines examined using western blotting.

**Figure 8 cancers-13-01747-f008:**
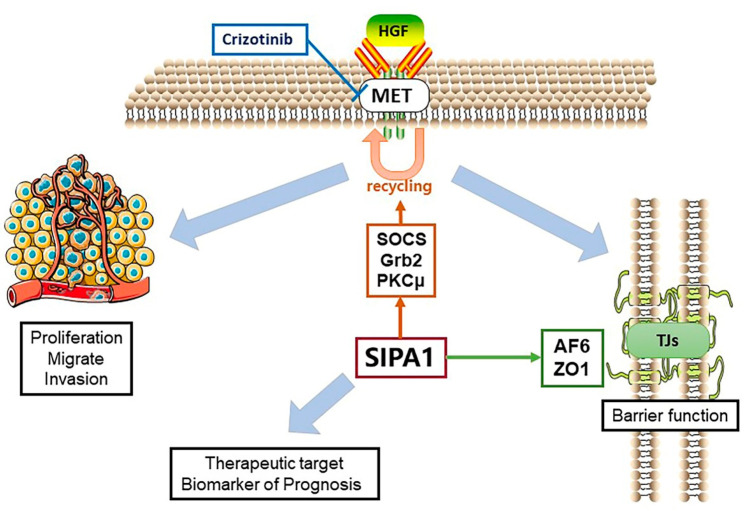
Diagram of the overview of this study. Schematic indicating the possible regulatory pathway for SIPA1 in lung cancer cells and the control of TJ function and assembly by HGF.

**Table 1 cancers-13-01747-t001:** Expression level of SIPA1 transcripts in relation to clinical and pathological demographics of the Peking lung cancer (and normal tissue) cohort.

	Parameters	Number of Samples	Median SIPA1 Expression (Copies/50 ng RNA)	Interquartile Range	
				Q1	Q3
Entire cohort	Tumour	148	25	12	74
	Normal	148	19	7	45
Entire cohort	Tumour	139	24.2	12	74
	Paired Normal	139	19	7	45
Histology type	Squamous Cell Carcinoma	50	25	14	62
	Adenocarcinoma	67	19.8	9.1	73.8
	Other	18	24	6	121
Degree of differentiation	High	7	14.88	3.05	21.88
	High to moderate	16	24.2	15.9	63.5
	Moderate	50	25	9	67
	Moderate to Low	23	26.9	12.2	96.7
	Low	14	17.9	7.5	42.5
T Staging	T-1	21	17.3	4.2	63
	T-2	57	21	9	65
	T-3	29	50	20	168
	T-4	16	18.3	6.1	65.3
Nodal staging	N-0	70	24.1	9	63.7
	N-1	22	31	11	82
	N-2	38	19	10	78
TNM staging	TNM1	39	17.3	6.9	59.7
	TNM2	31	52	20	189
	TNM3	51	19	12	63
	TNM4	1	0.1134	*	*
TNM sub-staging	TNM1A	17	15.5	4.1	63
	TNM1B	22	21.9	8.2	60.3
	TNM2A	2	135		*
	TNM2B	29	52	20	189
	TNM3A	37	20	14	63
	TNM3B	14	18.3	6.4	77.1
Smoking history	Non smoker	59	28.5	9.7	63.8
	Smoker	77	20	11	71

* denotes values too small to calculate.

## Data Availability

The data presented in this study are available on request from the corresponding author. The data are not publicly available due to ongoing research studies.

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
