# Peer review of "SIPA1 Is a Modulator of HGF/MET Induced Tumour Metastasis via the Regulation of Tight Junction-Based Cell to Cell Barrier Function"

_cancers, 2021, doi:10.3390/cancers13071747_

Round 1

Reviewer 1 Report

Authors described their finding in SIPA1 that is a modulator of HGF/MET-induced tumor metastasis via the regulation of tight junction-based cell to cell barrier function. Authors have shown very interesting results and implications. However, the manuscript has shown some concerning points listed below:  

Main concerns:

  • Many abbreviations used in the manuscript do not have full names in their first-time use, nor any description on their functions.
  • In introduction, it will be helpful to add a brief explanation on the conditions in which HGF (hepatocyte growth factor) is expressed so that HGP/MET signaling is triggered.
  • It is confusing that authors mentioned “Higher expression levels of SIPA1 were associated with worse prognosis and increased incidence of metastases for prostate cancer (CaP) patients” with an example of “SIPA1 also intensifies the invasion efficiency, but decreases the attachment ability, of CaP cells by down regulating BRD4 (bromodomain protein 4) and ECM (extra cellular matrix)-related gene expression.” Soon after, author described that “similar effect was observed in the cells with SIPA1 knocked down cells.” How does a similar effect can be observed from the same type of cells with either a higher expression or knocked down of SIPA1. If author meant to say similar effect in the cells with either a higher expression or knocked down of SIPA1 in different cancer types, the similar effect observed are due to tissue specificity? Are these cells share similar signaling pathway in tumorigenesis and metastasis?
  • In Fig. 4D, the backgrounds of microscopic images are noticeably different from each other and hardly justify the authors’ claim on the expression level of the indicated proteins. Better images with similar background with a standardized, statistic measurement on the expression of the proteins are required for the conclusion author would like to achieve.
  • The purpose and finding of comparing the phosphorylation status between A549 pEF-CT cells and the A549 SIPA1-KD cells 386 are not clear.
  • Tight junction functions to maintain permeability and polarity, prevents migration and motility. It appears conflicting that SIPA1 KD cells showed decreased tight junction-based barrier function by downregulating MET while they showed lower capability of migration and motility measured in Transwell assay.
  • In Fig. 6B, what happens to the cells at 15h of resistance experiment? The time point of 15h in the experiment appears to be a turning point for HGF signaling in TJ function between CT and KD cells. Any explanation?
  • In Fig. 6F, the variation of MET expression and phosphorylation status is significant. Thus, statistical measurements using GAPDH as the loading control to normalize the amount of MET and pMET is required to reach the authors’ conclusion. In addition, the molecular difference on the phosphorylated and non-phosphorylated MET need to be addressed.
  • In Fig. 7, the relevance between the proteins with the internalization and recycling of MET was not clear.
  • SIPA1 is a Rap GAP. Therefore, knocking down of SIPA1 is expected to result in higher Rap1 activation and subsequent stronger cell adhesion, consistent with less mobility/invasion as authors shown. What are the potential reasons of SIPA1 that is not considered to function as a Rap GAP in lung cancer cells?

Minor points:

  • A brief description of SKMES1, CORL23, and A549 cell lines is necessary to help reader to understand the purpose of authors to examine SIPA1’s function in these cell lines.
  • pEF-CT cells are assumed to be the control parent cells without knockdown and need to be explained.
  • In the section of “3.4. TJ markers were influenced by SIPA1 in lung cancer cells”, many abbreviations for the TJ protein family need their full names and brief description for their roles in TJ functions. The relevance of the upregulated or downregulated proteins in TJ function needs explanation.
  • Confusing sentences, such as “Marvel D3, binding protein belonging to the TAMP family…”, “The interaction of SIPA1 and HGF/MET signaling on the regulation of TJs in lung cancer cells.” “RT-PCR and qPCR showed that the transcript level of MET was not influenced by knockdown of SIPA1 in A549 lung cancer cells, and in SK-MES1 lung cancer cells, qPCR  showed the transcript level of MET was actually higher in SIPA1-KD cells compared to pEF-CT cells”
  • All genes mentioned in the sentence of “Correlation analysis on the TCGA-LUAD database also demonstrated that TJ markers such as the gene F11R (encoding JAM1 protein), the ZO family encoding genes, Claudin proteins encoding genes such as CLDN4, CLDN7, CLDN12, CLDN15, and Marvel D3” are actually proteins. If authors prefer to use gene names, they should use a standardized way to present them.
  • Repetitive use of word, such as “SIPA1 regulates MET at the protein level, by regulating the internalization and reuse of MET”
  • A consistency of the cell-line names in the different sections of manuscript.

Author Response

Many thanks to the reviewer for their constructive and helpful comments and suggestions. Our responses are outlined below, and the revised manuscript shows these changes as tracked.

Main concerns:

Many abbreviations used in the manuscript do not have full names in their first-time use, nor any description on their functions. These have now been added, thank you.

In introduction, it will be helpful to add a brief explanation on the conditions in which HGF (hepatocyte growth factor) is expressed so that HGP/MET signaling is triggered. This has been inserted. 57-59

It is confusing that authors mentioned “Higher expression levels of SIPA1 were associated with worse prognosis and increased incidence of metastases for prostate cancer (CaP) patients” with an example of “SIPA1 also intensifies the invasion efficiency, but decreases the attachment ability, of CaP cells by down regulating BRD4 (bromodomain protein 4) and ECM (extra cellular matrix)-related gene expression.” Soon after, author described that “similar effect was observed in the cells with SIPA1 knocked down cells.” How does a similar effect can be observed from the same type of cells with either a higher expression or knocked down of SIPA1. If author meant to say similar effect in the cells with either a higher expression or knocked down of SIPA1 in different cancer types, the similar effect observed are due to tissue specificity? Are these cells share similar signaling pathway in tumorigenesis and metastasis? This has been corrected in the text. 104-106

In Fig. 4D, the backgrounds of microscopic images are noticeably different from each other and hardly justify the authors’ claim on the expression level of the indicated proteins. Better images with similar background with a standardized, statistic measurement on the expression of the proteins are required for the conclusion author would like to achieve. These images are the original ones and we do not expect to have to “correct” them. We have explained in the text that it is the distribution that is the important change rather than the gross levels. 377-379

The purpose and finding of comparing the phosphorylation status between A549 pEF-CT cells and the A549 SIPA1-KD cells 386 are not clear. This has been added. 406-409

Tight junction functions to maintain permeability and polarity, prevents migration and motility. It appears conflicting that SIPA1 KD cells showed decreased tight junction-based barrier function by downregulating MET while they showed lower capability of migration and motility measured in Transwell assay. An interesting observation, thank you.

In Fig. 6B, what happens to the cells at 15h of resistance experiment? The time point of 15h in the experiment appears to be a turning point for HGF signaling in TJ function between CT and KD cells. Any explanation? We have now commented on this, thank you. 422-423

In Fig. 6F, the variation of MET expression and phosphorylation status is significant. Thus, statistical measurements using GAPDH as the loading control to normalize the amount of MET and pMET is required to reach the authors’ conclusion. In addition, the molecular difference on the phosphorylated and non-phosphorylated MET need to be addressed. We do not believe that statistical analysis will be of benefit here as it is evident that phosphorylation and total met is reduced. This has been commented on in the text. 439

In Fig. 7, the relevance between the proteins with the internalization and recycling of MET was not clear. This has been amended in the text. 451

SIPA1 is a Rap GAP. Therefore, knocking down of SIPA1 is expected to result in higher Rap1 activation and subsequent stronger cell adhesion, consistent with less mobility/invasion as authors shown. What are the potential reasons of SIPA1 that is not considered to function as a Rap GAP in lung cancer cells? Apologies, your question is not clear.

Minor points:

A brief description of SKMES1, CORL23, and A549 cell lines is necessary to help reader to understand the purpose of authors to examine SIPA1’s function in these cell lines. This has been stated in the materials and methods.

pEF-CT cells are assumed to be the control parent cells without knockdown and need to be explained.  This has now been stated in the methodology, thank you.

In the section of “3.4. TJ markers were influenced by SIPA1 in lung cancer cells”, many abbreviations for the TJ protein family need their full names and brief description for their roles in TJ functions. These have been inserted.

The relevance of the upregulated or downregulated proteins in TJ function needs explanation. The relevance of these findings would require a great deal of further work and are beyond the scope of the current manuscript and so we have commented on this.

Confusing sentences, such as “Marvel D3, binding protein belonging to the TAMP family…”, “The interaction of SIPA1 and HGF/MET signalling on the regulation of TJs in lung cancer cells.” “RTPCR and qPCR showed that the transcript level of MET was not influenced by knockdown of SIPA1 in A549 lung cancer cells, and in SK-MES1 lung cancer cells, qPCR showed the transcript level of MET was actually higher in SIPA1-KD cells compared to pEF-CT cells” These sentences have been corrected to be hopefully less confusing.

All genes mentioned in the sentence of “Correlation analysis on the TCGA-LUAD database also demonstrated that TJ markers such as the gene F11R (encoding JAM1 protein), the ZO family encoding genes, Claudin proteins encoding genes such as CLDN4, CLDN7, CLDN12, CLDN15, and Marvel D3” are actually proteins. If authors prefer to use gene names, they should use a standardized way to present them. These have been changed.

Repetitive use of word, such as “SIPA1 regulates MET at the protein level, by regulating the internalization and reuse of MET”. Apologies, however, this is an accurate description of the results we found and requires stating.

A consistency of the cell-line names in the different sections of manuscript. Thank you we believe that they are now consistent.

Reviewer 2 Report

Authors presents interesting finding about SIPA1 role in lung tumorigenesis and metastasis. Authors have used quantitative analysis at RNA and protein level to addresses the hypothesis of SIPA1 role in regulating the TJs and reacting to HGF signaling in lung cancer cells. The study has potentially interesting new data, but few minor comments are below.

  • Can authors explain why some of their data have borderline significance? As this study suggest this data can be used for therapeutics in cancer treatment and the challenge in the field is specificity.
  • Authors have not mentioned what "star: represent in table and figure? They should add it in the legend.
  • Authors have not mentioned about number of replicates for the figures representing statistical analysis. Please add number of replicates in legend for figures.

Author Response

Thank you to reviewer 2 for their kind comments and suggestions. Our responses are outlined below, and the revised manuscript shows these changes as tracked.

Can authors explain why some of their data have borderline significance?

Patient data often requires considerable numbers in order to achieve significance, hence why significance if borderline. Within in vitro work, cell lines do not always react in the same manner for numerous experiments as they are living systems and as such are subject to vagaries in behaviour. This is well known, if frustrating. No experiment is perfect.

As this study suggest this data can be used for therapeutics in cancer treatment and the challenge in the field is specificity.

This is a true challenge, specificity is always an issue and we agree with the reviewer.

Authors have not mentioned what "star: represent in table and figure?

They should add it in the legend.

This has now been added, thank you, it indicates significance.

Authors have not mentioned about number of replicates for the figures representing statistical analysis. Please add number of replicates in legend for figures."

This has also been added, n=3 is the minimum.

Reviewer 3 Report

General comments:

In this paper, Liu et al explored the role of SIPA1 in lung cancer by using the lung cancer tissue samples and the lung cancer cell lines, and claim that SIPA1 plays an essential role in non-small cell lung cancer tumorigenesis and metastasis by enhancing invasion and proliferation and suppressing the barrier function of lung cancer cells. Although the roles of SIPA1 in cell functions are complicated, the current findings are interesting and this study may provide some clues for further investigation on the role of SIPA1 in tumor cells. However, there are several major concerns, and there are many errors in description, which should be extensively revised.

Specific comments:

1) Fig. 4, lines 328-360: In regard to Fig. 4, the description seems to be incorrect in many aspects and should be carefully revised. For example:

Line 330: “ZO1 and ZO3 are apparently upregulated” (Fig. 4A), but not “ZO family were not significantly significant”.

Line 330: AF6 was “downregulated”, but not “upregulated”. (Fig. 4A)

Line 335: In addition to claudin 5, claudin 10, 15, 19, 20, 22, 23, and 24 appear to be “upregulated” at the transcriptional level (Fig. 4A). By contrast, claudin 5, 10, and 15 appear to be “decreased” at the protein level (Fig. 4C). Therefore, the findings are inconsistent. The explanation and the revision are required.

Lines 351-353: Although claudin 10 and 15 protein levels are “decreased” by knockdown of SIPA1 in A549 cells (Fig. 4C), both molecules are “upregulated” at the mRNA level (Fig. 4A). Therefore, the findings are inconsistent.

Line 359: “The intensity of ZO1 was strong in the pEF control cells but weak in the SIPA1 knockdown cells (Fig. 4D)”. By contrast, ZO1 is apparently upregulated at the transcriptional and the protein levels in the SIPA1 knockdown cells (Figs. 4A and 4C). Therefore, the findings are inconsistent. The explanation and the revision are required.

JAM family: The levels of JAM 1, 2, and 3 are not changed at the transcriptional level (Fig. 4A), but appear to be decreased at the protein level (Fig. 4C). Therefore, the findings are inconsistent. The explanation and the revision are required.

The symbols indicating the statistical significance should be clearly depicted in Figs. 4A and 4B.

2) Line 414: “The phosphorylation level of MET was downregulated by knockdown of SIPA1 in A549 lung cancer cell lines.” It appears that the decreased level of phosphorylated MET may just reflect the decreased level of total MET protein, but not specific reduction in phosphorylation of MET (Fig. 6F). The explanation is required.

Minor comments:

Lines 26, 478, 481, and 485: “SCOS” should be “SOCS”.

Line 261: “cancer from and” should be “cancer and”.

Line 425: “SOCS4” may be “SOCS5”.

Line 497: Claudin 10 is not downregulated (Fig. 4A).

Line 499: AF6 is downregulated, but not upregulated (Fig. 4A).

Line 547: “for of SIPA1” should be “for SIPA1”.

Author Response

We wish to thank the reviewer for their kind and constructive comments and suggestions. Our responses are outlined below, and the revised manuscript shows these changes as tracked.

Specific comments:

  • 4, lines 328-360: In regard to Fig. 4, the description seems to be incorrect in many aspects and should be carefully revised.

Line 330: “ZO1 and ZO3 are apparently upregulated” (Fig. 4A), but not “ZO family were not significantly significant”. This has been corrected.

Line 330: AF6 was “downregulated”, but not “upregulated”. (Fig. 4A). We have changed this, thank you.

Line 335: In addition to claudin 5, claudin 10, 15, 19, 20, 22, 23, and 24 appear to be “upregulated” at the transcriptional level (Fig. 4A). By contrast, claudin 5, 10, and 15 appear to be “decreased” at the protein level (Fig. 4C). Therefore, the findings are inconsistent. The explanation and the revision are required. An explanation can be read in the revised manuscript, lines 335 onwards.

Lines 351-353: Although claudin 10 and 15 protein levels are “decreased” by knockdown of SIPA1 in A549 cells (Fig. 4C), both molecules are “upregulated” at the mRNA level (Fig. 4A). Therefore, the findings are inconsistent. We have commented on this at this point in the manuscript. 337-360

Line 359: “The intensity of ZO1 was strong in the pEF control cells but weak in the SIPA1 knockdown cells (Fig. 4D)”. By contrast, ZO1 is apparently upregulated at the transcriptional and the protein levels in the SIPA1 knockdown cells (Figs. 4A and 4C). Therefore, the findings are inconsistent. The explanation and the revision are required. This has been revised. 377-379

JAM family: The levels of JAM 1, 2, and 3 are not changed at the transcriptional level (Fig. 4A), but appear to be decreased at the protein level (Fig. 4C). Therefore, the findings are inconsistent. The explanation and the revision are required. We have explained this. Section 3.4

The symbols indicating the statistical significance should be clearly depicted in Figs. 4A and 4B. Thank you, however, we believe that larger symbols will obscure the graph.

  • Line 414: “The phosphorylation level of MET was downregulated by knockdown of SIPA1 in A549 lung cancer cell lines.” It appears that the decreased level of phosphorylated MET may just reflect the decreased level of total MET protein, but not specific reduction in phosphorylation of MET (Fig. 6F). The explanation is required. We have commented on this. 446

Minor comments:

Lines 26, 478, 481, and 485: “SCOS” should be “SOCS”. This has been corrected.

Line 261: “cancer from and” should be “cancer and”. This has been corrected.

Line 425: “SOCS4” may be “SOCS5”. This has been corrected.

Line 497: Claudin 10 is not downregulated (Fig. 4A). This has been corrected.

Line 499: AF6 is downregulated, but not upregulated (Fig. 4A). Line 547: “for of SIPA1” should be “for SIPA1”. This has been corrected.

Round 2

Reviewer 1 Report

Authors have made a good improvements during last revision. However, some major concerns still remain.

Major concerns:

  1. About fig. 4b, I believe the images are original as authors stated. However, such significant difference in the background of the images indicates the acquisition conditions for these images are not identical. Author need to show images acquired with the exact same acquisition conditions and then quantify the results.
  2. About Fig. 6F, authors misunderstand the relationship between expression level and phosphorylation status of a protein. That is the exact reason statistical analysis is crucial to reach authors’ claim and conclusion. In addition, author have not explained the reason for the molecular change in the indicated protein.
  3. SIPA1 is a Rap GAP. Therefore, knocking down of SIPA1 is expected to result in higher Rap1 activation and subsequent stronger cell adhesion, consistent with less mobility/invasion as authors shown, indicating that SIPA1 does function as a RapGAP in the cell lines used the present study. It needs to be discussed.
  4. Please provide explanation on the question “Tight junction functions to maintain permeability and polarity, prevents migration and motility. It appears conflicting that SIPA1 KD cells showed decreased tight junction-based barrier function by downregulating MET while they showed lower capability of migration and motility measured in Transwell assay.”

Author Response

We thank the reviewer for their comments and answer as follows:

“Authors have made a good improvements during last revision. However, some major concerns still remain.”

Major concerns:

  1. About fig. 4b, I believe the images are original as authors stated. However, such significant difference in the background of the images indicates the acquisition conditions for these images are not identical. Author need to show images acquired with the exact same acquisition conditions and then quantify the results.

This has been amended and inserted into Fig 4.

  1. About Fig. 6F, authors misunderstand the relationship between expression level and phosphorylation status of a protein. That is the exact reason statistical analysis is crucial to reach authors’ claim and conclusion. In addition, author have not explained the reason for the molecular change in the indicated protein.

Respectfully, we do understand the relationship. This has been explained.

  1. SIPA1 is a Rap GAP. Therefore, knocking down of SIPA1 is expected to result in higher Rap1 activation and subsequent stronger cell adhesion, consistent with less mobility/invasion as authors shown, indicating that SIPA1 does function as a RapGAP in the cell lines used the present study. It needs to be discussed.

Whilst SIPA1 is a RAP GAP, we are describing here it’s downstream effects on TJ formation and function. The other reviewers are satisfied with our explanations and interpretations.

  1. Please provide explanation on the question “Tight junction functions to maintain permeability and polarity, prevents migration and motility. It appears conflicting that SIPA1 KD cells showed decreased tight junction-based barrier function by downregulating MET while they showed lower capability of migration and motility measured in Transwell assay.”

You appear to be mistaken, SIPA1-KD cells cause an increase in barrier function (section 3.3), which is consistent with the reduction in MET and reduced migratory capacity.

Reviewer 3 Report

This paper is appropriately revised in response to the comments.

Author Response

Thank you very much for your suggestion.